# Yin Yang 1: Role in Leishmaniasis

**DOI:** 10.3390/cells14151149

**Published:** 2025-07-25

**Authors:** Devki Nandan, Dilraj Kaur Longowal, Neil Reiner

**Affiliations:** Division of Infectious Diseases, Department of Medicine, University of British Columbia, Vancouver, BC V6T 1Z4, Canada; dilrajkaur.longowal@ubc.ca (D.K.L.); nreiner@mail.ubc.ca (N.R.)

**Keywords:** Yin Yang 1, leishmaniasis, host–pathogen interactions, parasitic infections, macrophages, transcription factors

## Abstract

Leishmaniasis, caused by protozoan parasites of the genus *Leishmania*, is one of the most neglected human diseases, affecting millions worldwide. A detailed understanding of the molecular mechanisms that govern the outcome of macrophage–*Leishmania* interactions is crucial for a comprehensive understanding of leishmaniasis; however, our current knowledge of these mechanisms remains limited. It is clear that *Leishmania* has co-evolved to engage several clever strategies to regulate the cell biology of host macrophages to survive and multiply in phagolysosomes of these cells. In this review, we discuss how *Leishmania* exploits the macrophage Yin-Yang 1 protein as a critical proxy virulence factor to promote its survival. Additionally, we discuss an atlas of YY1-dependent proteins in human macrophages, which could serve as a valuable resource for researchers studying the role of YY1 in macrophage cell biology.

## 1. Introduction

The protozoan parasites of the genus *Leishmania* cause the human parasitic disease known as leishmaniasis. Leishmaniasis encompasses a spectrum of diseases with three primary clinical manifestations, cutaneous, mucocutaneous, and visceral, all of which affect millions of people around the globe [1,2,3]. Unfortunately, leishmaniasis is still one of the neglected diseases, occurring endemically in tropical, subtropical and Mediterranean regions of the world [3,4]. Its incidence is increasing due to the absence of approved human vaccines, the use of ineffective and highly toxic drugs, the emergence of drug resistance, rising tourism, armed conflicts in affected areas, and global warming [5,6]. Species of the genus *Leishmania* have a complex dimorphic biological life cycle. A non-motile amastigote form resides in mammalian hosts, including humans, while a motile promastigote form resides in an invertebrate vector (phlebotomine sand flies). The amastigote form primarily colonizes phagolysosomes of macrophages and is responsible for the pathophysiology of all forms of leishmaniasis in humans. Ironically, macrophages, the primary immune cells responsible for eliminating *Leishmania*, also provide a haven for its persistence.

Macrophage–*Leishmania* interactions directly influence the outcome of the disease. Clearly, *Leishmania* has co-evolved with the host to manipulate host cell biology for its survival and multiplication. In-depth knowledge of the molecular mechanisms of the macrophage–*Leishmania* antagonistic relationship could pave the way to developing effective anti-leishmanial therapeutic strategies.

The functional control of macrophages occurs at multiple levels, such as transcriptional regulation [7], the involvement of RNA-binding proteins [8], miRNA-mediated decay [9], metabolic rewiring, and post-translational modifications [10]. It appears that *Leishmania* primarily controls macrophage functions at the transcriptional level by hijacking host cell signaling pathways. The involvement of various transcription factors in regulating the immune-metabolic response of *Leishmania*-infected macrophages has recently been reviewed [11]. For example, signal transducers and activators of transcription (STATs), such as STAT1, interferon regulatory factors (IRFs), nuclear factor kappa B (NF-κB), and hypoxia-inducible factor (HIF-α), are involved in the pro-inflammatory M1 polarization of macrophages during infection. Additionally, IRF4, STAT6, and peroxisome proliferator-activated receptor (PPAR), which are induced by IL-4, regulate M2 gene expression. It has also been demonstrated that *Leishmania* targets the macrophage transcription factors c-Myc and Yin-Yang 1 (YY1) as virulence factors by proxy to promote survival within infected cells [12,13]. The role of c-Myc in the survival of *Leishmania* inside macrophages has recently been highlighted [14]. In this review, we will emphasize the complex role of macrophage YY1 in the progression of *L. donovani* infection in macrophages. We will also overview the current understanding of the mechanistic and functional properties of YY1, as well as the macrophage genes affected by YY1 knockdown using siRNAs. Together, this review will add a novel function to the growing list of roles assigned to YY1 and unveil its importance in the field of infection and immunity.

## 2. Overview of the Structure and Function of YY1

YY1 is a multifaceted and ubiquitous protein that plays a role in diverse biological functions of eukaryotic cells, including cell growth [15], proliferation [16], differentiation [17], cell cycle [15], DNA repair [18], and apoptosis [19,20]. However, the specific mechanisms by which YY1 executes these functions remain unclear. It is estimated that YY1 regulates the transcription of approximately 7–10% of mammalian genes [21,22]. The human YY1 gene is located on chromosome 14q32.2, and the cDNA sequence predicts a molecular weight of 44 kDa (414 amino acids). However, the SDS-PAGE-based molecular weight of YY1 is 65–68 kDa, probably due to extensive post-translational modifications. This protein was independently discovered by various groups in 1991, who gave different names to YY1 based on the cellular context and molecular mechanisms associated with it. It was referred to as YY1 by Shi et al. [23], nuclear factor E1 (NF-E1) by Park and Atchison [24], and delta (δ) by Hariharan et al. [25]. Later, YY1 was further referred to as upstream conserved region binding protein [26], nuclear matrix protein 1 [27], nuclear factor D (NF-D), or F-ACT1 [27]. The current name of Yin Yang 1 is due to its bifunctionality as a transcription activator and repressor.

It has been reported that a large number of genes possess potential YY1 binding sites, many of which have been experimentally validated, including those in c-Fos, c-Myc, and histones H3.2, H4 [23,28,29]. YY1 belongs to the GLI-Kruppel-like family of proteins, which contains zinc-finger domains with dual transcriptional activities. Depending on the interacting partners and cellular context, YY1 can regulate gene expression through various direct or indirect mechanisms: 1. It serves as a traditional DNA-binding transcription factor. Interestingly, the YY1 promoter lacks a classical TATA box region; however, it contains transcription factor Sp-1 sites and a region enriched in CG dinucleotides, resembling promoters found in essential housekeeping genes, which suggests YY1’s involvement in crucial biological processes, including development [30]. 2. It acts as a synergistic enhancer via protein–protein interactions. 3. YY1 could also function as an architectural protein that regulates transcription by facilitating enhancer–promoter interactions. Due to its dimerization ability, it loops two distant elements in the genome, mediating the interaction between these sequences.

As expected from the multifunctional YY1 protein, it contains several domains, as schematically represented in Figure 1 [31]. The N terminus of YY1 (1–155 aa) is a transactivation domain with distinctive features. It interacts with positively charged proteins due to eleven acidic residues between 43 and 53 amino acids [20]. This acidic track is followed by the histidine track of 11 residues, which appears to be involved in YY1 localization [32]. A region rich in glutamine and proline (aa 81–100) is also present and is referred to as the second acidic track [33]. The glycine/alanine-rich central region (170–226 aa) and glycine/lysine-rich sequence (333–397) overlap with the zinc fingers near the C-terminus, which have been implicated in the repressive capacity of YY1. The C-terminus of YY1 contains four C2H2 zinc-finger motifs required for its DNA-binding capability [34,35]. In addition, it is also becoming apparent that zinc-finger motifs are involved in protein–protein interactions [36]. YY1 also possesses a domain known as REPO domain (from aa 201 to 226) due to its capability to recruit the Polycomb group of proteins to DNA [37,38]. Recently, this domain has also been referred to as the OncoProtein Binding (OPB) domain, which is required for oligomerization and interaction with oncoproteins [39].

## 3. Regulation of YY1 Activity

Currently, it is not known what governs the rules of activation/repression activity of YY1. A variety of molecular mechanisms have been implicated in regulating the function of YY1, including its associated co-factors, subcellular localization, and post-transcriptional and post-translational modifications. Regulation of YY1 activity at the post-transcriptional and post-translational modifications level is briefly described below. It is known that YY1 activity can be regulated at the post-transcriptional level through alternative splicing and by affecting mRNA stability. RNA splicing can influence YY1 activity by generating different mRNA isoforms, which may modify YY1’s function [40]. For example, different YY1 mRNA isoforms may vary in their ability to bind specific DNA sequences or interact with other transcription-related proteins, causing changes in target gene expression and affecting various cellular processes. YY1 activity is also regulated by the stability of its mRNA, which is influenced by various factors, including modifications like m6A methylation and interactions with other molecules. These interactions can either stabilize or destabilize YY1 mRNA, thereby affecting its levels and ultimately impacting cellular functions. For instance, METTL3-mediated m6A modifications increase YY1 mRNA stability, promoting faster growth in multiple myeloma cells [41]. On the other hand, the enzyme ALKBH5, which removes m6A modifications, can decrease YY1 mRNA stability, as seen in gastric cancer cells [42].

YY1 activity can also be regulated at the post-translational level. It is well established that post-translational modifications (PTMs) play a crucial role in altering protein conformation, activity, stability, localization, and interactions with DNA, RNA, and other proteins, thereby influencing the phenotypes and biological processes of cells. Abnormal PTMs can cause changes in protein properties and a loss of protein functions, which are closely associated with the development of various diseases [43,44,45,46,47]. The most common types of PTMs on YY1 include phosphorylation [35], glycosylation [48], methylation [49], ubiquitination [50], SUMOylation [51], acetylation [52], and redox modifications [53]. Gaining insight into these modifications is key to understanding the multiple roles of YY1 in different normal cellular processes and disease conditions. Creating a multi-omics molecular network that encompasses the genome, proteome, metabolome, and PTMome will facilitate new breakthroughs in identifying drug targets against leishmaniasis [54,55].

It is worth noting that several studies have shown the involvement of long non-coding RNAs (lncRNAs) in YY1’s regulatory network. YY1 can regulate lncRNAs and lncRNAs can also act as regulatory molecules for YY1 itself. A recent discussion has focused on the mechanisms of YY1 and lncRNA interactions during tumor progression [56]. Additionally, lncRNAs can modulate protein localization either directly or indirectly. This includes proteins involved in the nuclear transport machinery, which either promote or inhibit the movement of proteins between the cytoplasm and the nucleus. It is tempting to speculate that lncRNAs are involved in the translocation of YY1 from the nucleus to the cytoplasm.

## 4. Subcellular Localization of YY1

YY1 is primarily localized to the nucleus, where it functions as a transcription factor. It has been shown that YY1 is associated with the nuclear matrix when it is not bound to DNA. In addition, YY1 is also found in the cytoplasm. Strikingly, YY1 localization alters during the cell cycle. YY1 is mainly cytoplasmic at the G1 phase, whereas it is primarily nuclear in the early and middle S phase and returns to the cytoplasm during the late S phase [57]. Interestingly, YY1 localization changes (cytoplasm to nucleus) during the induction of apoptosis [58]. Furthermore, pathogens can also trigger the movement of YY1. For example, the vaccinia virus causes YY1 to relocate from the nucleus to the cytoplasm in infected human macrophages, playing a role in the virus’s pathogenesis [59,60].

## 5. YY1 Protein Relevance to Leishmaniasis

Several known functions of YY1 that could impact the pathophysiology of leishmaniasis, likely favoring *Leishmania* persistence in host macrophages. For example, YY1 is involved in M2 macrophage polarization, a recognized immune-suppressive and anti-inflammatory subtype; it likely creates an immune-suppressive environment that enables *Leishmania* persistence [61,62,63,64]. Additionally, YY1 has been shown to directly activate the cell signaling protein Akt [65] in breast cancer cells. In a study related to leishmaniasis, it was shown that *L. donovani* also activates Akt to promote its intramacrophage survival, seemingly by regulating the GSK-3β/β-catenin/FOXO-1 axis, which leads to the attenuation of both host cell apoptosis and immune responses essential for parasite survival [66]. In another study, it was revealed that *L. donovani* induced Akt phosphorylation in a PI3K-dependent manner, resulting in the inactivation of glycogen synthase kinase-3β, followed by an increase in the production of IL-10, thus creating an immune-suppressive environment suitable for *Leishmania* growth [67]. Furthermore, Akt inhibition directly resulted in a significant decrease in parasite survival [68]. YY1 also interacts with STAT1 in the cytoplasm, leading to a reduction in STAT1-activated genes [69]. Previous studies have demonstrated that *Leishmania*, similar to YY1, inhibits STAT-1-mediated cell signaling in host macrophages to evade host defense mechanisms and promote its survival [70]. A recent transcriptomic profiling study identified YY1 as one of the transcriptional regulators activated in response to *Leishmania* infection in macrophages, suggesting YY1’s potential involvement in enhancing *Leishmania* survival [71]. As discussed above, vaccinia virus can trigger the movement of YY1 from the nucleus to the cytoplasm in infected human macrophages, thereby playing a role in the virus’s pathogenesis [59,60]. This suggests that *Leishmania* may also prompt YY1 to move to the cytoplasm, enabling its interaction with Akt and STAT1, which could create a host environment conducive to *Leishmania* survival. Together, this generated a hypothesis that YY1 is a multifunctional protein likely to impact macrophage biology in *Leishmania* infection, favoring its survival. The intent of this review is not to discuss in detail the above potential relationship of YY1 to leishmaniasis. This review will primarily focus on newer studies that strongly suggest a role for the host macrophage YY1 in the persistence of *Leishmania.*

### 5.1. Role of YY1 in Leishmania Perseverance

As discussed above, several functions of YY1 could contribute to the pathobiology of leishmaniasis. A recent study [13] from our group has directly addressed the role of the host human macrophage YY1 protein in the survival of *L. donovani* in infected cells. This study also evaluated the subcellular movement of YY1 in infected cells, asking whether, like Vaccinia virus, *Leishmania* triggers the transition of YY1 from the nucleus to the cytoplasm. Characterization of the macrophage proteome in conditions of low YY1 abundance in infected cells revealed YY1-dependent macrophage proteins sensitive to *Leishmania* infection. Beyond infection-related investigation, this study also characterized the global proteome of macrophages dependent on the optimal concentration of YY1. This review mainly focuses on discussing this newer study.

### 5.2. Regulation of Macrophage YY1 Protein by Leishmania

Brar et al. [13] explored the role of macrophage YY1 protein in *Leishmania* infection. For this investigation, they employed siRNA-mediated downregulation of macrophage YY1 for 24 h before infection. The *Leishmania* survival assay was used to determine whether YY1 confers a survival advantage to *Leishmania*. Interestingly, downregulation of host YY1 resulted in the attenuation of *Leishmania* survival in differentiated THP-1 cells. These findings were validated in human monocyte-derived macrophages (Figure 2A,B). Taken together, these results indicate that YY1 plays a critical role in enabling *L. donovani* to survive within macrophages.

The role of YY1 in *Leishmania* infection is expected to be complex, as it is involved in the expression of a large number of proteins. To understand the regulation of YY1 protein by *Leishmania*, Brar et al. [13] investigated the total abundance of YY1 protein in infected cells compared to non-infected macrophages through a Western blot assay. They did not observe any significant modulation of YY1 abundance at the whole-cell level. Cellular fractionation studies revealed that *Leishmania* infection translocated YY1 from the nucleus, where it is primarily located, to the cytoplasm. In the context of host–pathogen interactions, an earlier study indicated that vaccinia virus triggered Crm1-dependent cytoplasmic translocation of YY1 within infected BSC-40 cells and human adherent monocytes [59,60], without altering the abundance of YY1 at the whole-cell level.

The increase in cytoplasmic YY1 levels could be the result of decreased nuclear import or increased YY1 nuclear export. To investigate this, the nuclear export of proteins was selectively inhibited using leptomycin B. Leptomycin B is known to inhibit the classic nuclear export nuclear pore complexes involving exportins (also known as chromosomal maintenance 1, Crm1) [72,73]. Western blotting (Figure 3) was used to assess the presence of YY1 in cytosolic fractions from both uninfected and *Leishmania*-infected cells. Leptomycin B treatment clearly led to reduced levels of cytoplasmic YY1 in both uninfected and infected cells, suggesting that cytoplasmic YY1 primarily originates from the export of nuclear YY1, possibly through the Crm1 system. This also ruled out the possibility of increased retention of newly produced cytoplasmic YY1 or decreased nuclear import. As expected, leptomycin treatment considerably reduced *Leishmania* survival in infected cells compared to those treated with the vehicle. This study does not identify the specific role of cytoplasmic YY1 in *Leishmania*-infected cells. However, the attenuated survival of *Leishmania* following leptomycin B treatment suggests that cytoplasmic YY1 may play a vital role in its survival. Discovering the interacting partners of YY1 in the cytoplasm of infected host macrophages could reveal the cytoplasmic function of YY1 in the pathobiology of *Leishmania* infection. The transition of YY1 to the cytoplasm may be necessary for its interaction with host factors, thus creating a favorable environment for *Leishmania’s* survival.

### 5.3. Characterization of the Global Proteome Reveals Host Macrophage Proteins Modulated by Leishmania and Dependent on YY1

This study investigated the YY1-dependent proteome of host macrophages during *L. donovani* infection. It involved a comprehensive and unbiased label-free quantitative proteomic analysis of *Leishmania*-infected YY1 knockdown. Of the total 6585 proteins identified, 252 were significantly affected by *L. donovani*, with 140 proteins upregulated and 112 downregulated. A comparative quantitative proteomic analysis was conducted between *L. donovani*-infected cells under reduced YY1 conditions and those in normal conditions. It was found that 31 of 245 proteins were significantly influenced by YY1 knockdown in infected cells, with 16 proteins recovering (eight upregulated and eight downregulated) as a result of the knockdown.

Furthermore, the host proteins sensitive to *L. donovani* infection and dependent on YY1 were categorized into three groups based on Gene Ontology (GO) terms: biological process, molecular function, and cellular component. GO term analysis of *Leishmania* upregulated YY1-dependent proteins revealed the top GO terms for biological processes as signaling (GO:0023052), cell differentiation (GO:0030154), and anatomical structure development (GO:0048856). GO term analysis of *Leishmania* downregulated YY1-dependent proteins revealed the top GO terms for biological processes as cell differentiation (GO:0030154) and protein-containing complex assembly (GO:0065003).

In addition to characterizing the infection-related macrophage proteome, this study revealed that approximately 5% of the identified proteins are sensitive to the abundance of YY1. To date, there is no published report describing the YY1-dependent proteome in human and murine macrophages. The following section provides a detailed discussion of YY1-sensitive macrophage proteome.

### 5.4. Comparative Proteome Analysis of YY1 Knockdown and Normal THP-1 Cells

Among the 6559 proteins identified across three replicates, 537 showed significant changes in abundance in the YY1 knockdown cells. Of these, 291 proteins were upregulated, whereas 246 were downregulated in the YY1 knockdown THP-1 cells. All significantly modulated proteins were categorized into three groups based on GO terms: biological process, molecular function, and cellular component. The most significant number of proteins modulated by YY1 knockdown were linked to anatomical structural development, cell differentiation, and signaling. A substantial number of proteins downregulated by YY1 knockdown were also involved in regulating DNA-templated transcription. Proteins affected by YY1 knockdown also played a role in programmed cell death. Regarding molecular function, both upregulated and downregulated proteins were primarily involved in catalytic activity. As for cellular localization, most downregulated or upregulated proteins were detected in both the nucleus and cytoplasm, with the highest concentration in the nucleus, as expected from YY1 being a transcription factor. A significant number of upregulated proteins were also identified as part of the plasma membrane.

This study also explored the significance of YY1-dependent macrophage proteins beyond leishmaniasis, particularly in the context of infection and immunity. To achieve this, a literature review was conducted on the role of YY1 in relation to infection and immunity, primarily utilizing the PubMed, Scopus, and Google Scholar databases for biological research. COVID-19 was excluded from this search. Of the 537 YY1-dependent proteins, 54 have been previously reported in infections and 50 in immunity-related studies. Notably, proteins upregulated by YY1 knockdown are more involved in infection and immunity.

Based on the studies discussed in this review, we propose a working model for the potential role of YY1 in *Leishmania* perseverance (Figure 4). *Leishmania* induces translocation of the host YY1 protein from the nucleus to the cytoplasm, promoting its interaction with STAT1 and AKT, thereby creating a pro-parasitic environment that allows *Leishmania* to survive and thrive in the hostile conditions within host macrophages. The purpose of this model is to stimulate thought and experimentation; it is not intended to be the sole mechanism for YY1-mediated creation of a pro-parasitic environment. This model also considers the potential role of YY1 PTMs in enhancing *Leishmania* persistence.

## 6. Concluding Remarks

This review summarizes the current knowledge of the role of YY1 in promoting *Leishmania* infection in human macrophages. During infection, *Leishmania* modulates several host macrophage proteins, employing YY1, which may be to its advantage. The *Leishmania*-mediated translocation of macrophage YY1 from the nucleus to the cytoplasm is particularly interesting, highlighting the multifunctional nature of YY1 and warranting further investigation. We also briefly discuss the expression of macrophage proteins, which depends on the level of YY1. In conclusion, this work makes a significant contribution to understanding macrophage–*Leishmania* interactions and host–pathogen interactions in general.

## 7. Unresolved Outstanding Questions

-This study has evaluated the role of macrophage YY1 in *Leishmania’s* optimal survival. However, it is not known how *Leishmania* regulates YY1 to ensure its survival. As discussed in Section 3, post-translational modifications such as phosphorylation, acetylation, and others influence YY1 function. It would be interesting to examine the potential role of PTMs of YY1 during *Leishmania* infection.-Although it is clear that *Leishmania* induces the translocation of YY1 from its primary location in the nucleus to the cytoplasm of infected cells, the mechanism behind this process remains unknown, as does the question of whether *Leishmania* molecules contribute to triggering the translocation.-This study was conducted using monocytic THP-1 cells and a single species of *Leishmania* (*L*. *donovani*). It will be interesting to validate the key findings of this study using primary human macrophages. Using other *Leishmania* species, such as *L. major* and *L. mexicana*, which result in different disease manifestations, will shed light on whether the regulation of YY1 is a central strategy utilized by other *Leishmania* species.

Addressing these outstanding questions will increase our understanding of the role of this seemingly crucial macrophage transcription factor in *Leishmania* infection and overall host–pathogen interactions.

## Figures and Tables

**Figure 1 cells-14-01149-f001:**
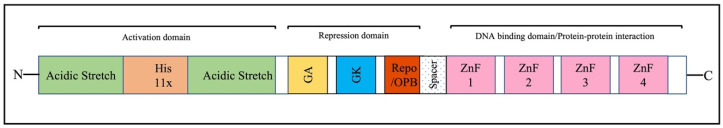
Schematic diagram of domains of the human YY1 protein.

**Figure 2 cells-14-01149-f002:**
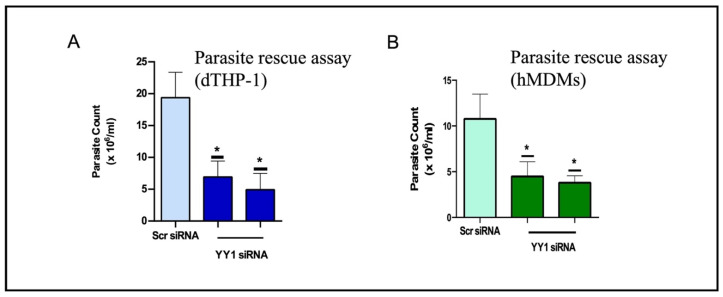
YY1 protein knockdown attenuates *Leishmania* survival in infected (**A**) THP-1 cells and (**B**) human monocyte-derived macrophages (hMDMs). (*: *p* < 0.05) This figure is modified from a published study [13].

**Figure 3 cells-14-01149-f003:**
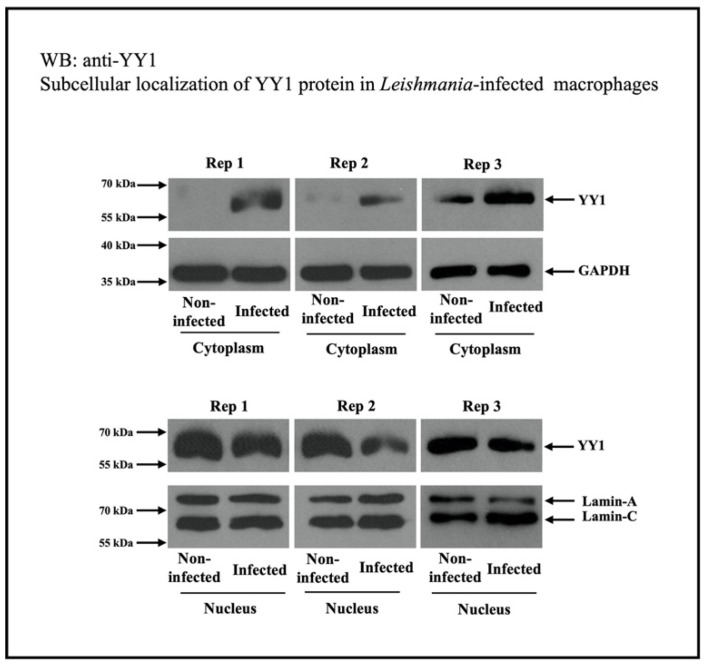
*Leishmania* infection triggers the cytoplasmic location of YY1 protein from the nucleus in THP-1 cells. The cytoplasmic and nuclear fractions were analyzed for the level of YY1. Three independent replicates are shown. This figure is modified from a published study [13].

**Figure 4 cells-14-01149-f004:**
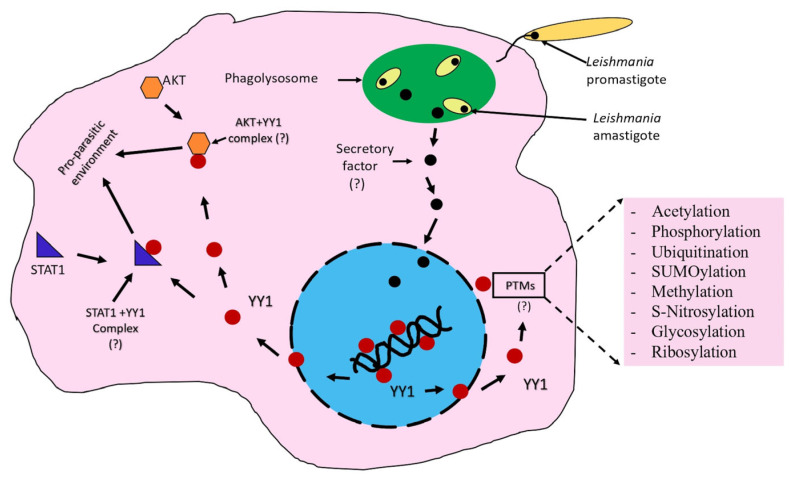
*Leishmania* redirects YY1 in infected macrophages. This model illustrates how *Leishmania* induces the translocation of YY1 from the nucleus (blue) into the cytoplasm (pink), allowing it to interact with Akt and STAT1, thereby creating a pro-parasitic environment. A potential mechanism for *Leishmania*-mediated YY1 translocation into the cytoplasm, involving *Leishmania* secretory factors, is depicted.

## Data Availability

This review summarizes the data that has already been published.

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
