# Peer review of "Yin Yang 1: Role in Leishmaniasis"

_cells, 2025, doi:10.3390/cells14151149_

Round 1
Reviewer 1 Report
Comments and Suggestions for Authors
It is a well-written review that will interest parasitologists and biochemists. I would recommend the authors to address the following comments.
Lines 42-43 claim that the functional control of macrophages primarily occurs at the transcriptional level. In reality, RNA-binding proteins, miRNA‑mediated decay, metabolic rewiring and PTMs such as phosphorylation are equally decisive. A short paragraph acknowledging post‑transcriptional and post‑translational layers would prevent the current oversimplification.
Reference 8 already provides a detailed overview of several transcription factors that regulate immune-metabolic responses in Leishmania infected macrophages. If would be more beneficial for the reader if the authors could explain in the introduction section what exactly the YY1-centric review would add.
Please make it clear to the reader that you are citing your previous work (reference 10) when you describe the proteomics study line 220 onward. This is not very obvious in the way the text is written at the moment.
Although figure 4 nicely describe your hypothesis, it still stops short of translational angle. Although it is up to the authors, a walk‑through of YY1 phosphorylation, acetylation, SUMOylation, etc. will be useful. Alternatively, a table summarising each PTM, its known effect on YY1 activity and any infection links would make it more comprehensive. You could also enrich the section 5 by including key outstanding questions on post‑transcriptional and post‑translational aspects.
Overall, the review tackles the emerging idea that host transcription factor YY1 is hijacked by Leishmania, nicely linking classic macrophage biology with new proteomic data. As suggested above, I believe that including discussion of post-transcriptional and post-translational regulatory mechanisms alongside transcriptional control would provide a more comprehensive overview of macrophage modulation during Leishmania infection.
Author Response
Concern 1: Lines 42-43 claim that the functional control of macrophages primarily occurs at the transcriptional level. In reality, RNA-binding proteins, miRNA‑mediated decay, metabolic rewiring and PTMs such as phosphorylation are equally decisive. A short paragraph acknowledging post‑transcriptional and post‑translational layers would prevent the current oversimplification.
Answer: We would like to thank this reviewer for finding our manuscript easy to read and interesting in the field of parasitology and biochemistry of host-pathogen interactions. Regarding concern number 1, we would like to point out that our intent was not to discuss the regulation of macrophage functions per se. It was mainly focused on cell-signalling-mediated regulation of macrophage function involving transcription factors. We have revised this section for clarity and added a new section (Section 3) discussing the regulation of YY1 at various levels to make it comprehensive.
Concern 2: Reference 8 already provides a detailed overview of several transcription factors that regulate immune-metabolic responses in Leishmania infected macrophages. If would be more beneficial for the reader if the authors could explain in the introduction section what exactly the YY1-centric review would add.
Answer: Here, we intended to give a brief discussion of transcription factors involved; however, as suggested by this reviewer, we have revised this section to clarify the objective of this review in the last paragraph of the introduction.
Concern 3: Please make it clear to the reader that you are citing your previous work (reference 10) when you describe the proteomics study line 220 onward. This is not very obvious in the way the text is written at the moment.
Answer: As suggested, we have addressed this concern by clearly stating that work cited in now reference # 13 is from our group.
Concern 4: Although figure 4 nicely describe your hypothesis, it still stops short of translational angle. Although it is up to the authors, a walk‑through of YY1 phosphorylation, acetylation, SUMOylation, etc. will be useful. Alternatively, a table summarising each PTM, its known effect on YY1 activity and any infection links would make it more comprehensive. You could also enrich the section 5 by including key outstanding questions on post‑transcriptional and post‑translational aspects.
Answer: We thank this reviewer for their suggestions to improve the message in model figure 4 by including potential modulation of YY1 PTMs during Leishmania infection. Accordingly, we have revised figure 4 to add a table of possible PTMs of YY1. We have also revised the section titled 'Key Outstanding Question' to include the potential role of PTMs, quoting verbatim: “As discussed in Section 3, post-translational modifications such as phosphorylation, acetylation, and others influence YY1 function. It would be interesting to examine the effect of these post-translational modifications of YY1 during Leishmania infection.”
Concern 5: Overall, the review tackles the emerging idea that host transcription factor YY1 is hijacked by Leishmania, nicely linking classic macrophage biology with new proteomic data. As suggested above, I believe that including discussion of post-transcriptional and post-translational regulatory mechanisms alongside transcriptional control would provide a more comprehensive overview of macrophage modulation during Leishmania infection.
Answer: As noted in the response to concern 1, the intent of this review was not to discuss macrophage regulation in detail; however, we have added a paragraph discussing the modulation of macrophage YY1 function at various levels.
Reviewer 2 Report
Comments and Suggestions for Authors
This review is interesting and well written, pointing on the proven role of Yin Yang 1 in leishmaniasis, whose future mechanism understanding will bring important knowledge.
May be the authors could remember that it would be important to know the possible association of Yin Yang with some long non coding RNA (lncRNAs), which for instance might be involved in the cytoplasmic import of Yin Yang in Leishmania infected macrophages.
That is to elaborate a model for the potential role of YY1 in Leishmania perseverance it could help to compare not only the transcribed RNAs between YY1 down regulated and not, macrophages and then Leishmania infected and not, but also the lncRNAs.
Line 174 you miss the full stop.

Author Response
Concern 1: May be the authors could remember that it would be important to know the possible association of Ying Yang with some long non coding (lncRNAs), which for instance might be involved in the cytoplasmic import of Ying Yang in Leishmania infected macrophages.
Answer: We appreciate this reviewer for critically examining our manuscript and highlighting the potential role of host lncRNAs in influencing YY1 function in Leishmania-infected macrophages. Accordingly, we have added a paragraph discussing this possibility in the last paragraph of Section 3.
Concern 2: That is to elaborate a model for the potential role of YY1 in Leishmania perseverance it could help to compare not only the transcribed RNA between YY1 down regulated and not, macrophages and then Leishmania infected and not, but also the lncRNAs.
Answer: This suggestion is regarding the inclusion of host lncRNAs in the regulation of YY1 in infected cells. As mentioned in response to concern 1 above, we have included this possibility in the last paragraph of Section 3. However, we found it challenging to include the possible role of lncRNAs in modulating YY1 function in the model without making it overly complicated.
Concern 3: Line 174 you miss the full stop.
Answer: We’ve corrected this typo.
Reviewer 3 Report
Comments and Suggestions for Authors
This is a timely and well-organized review that summarizes current knowledge on the role of the transcription factor Yin Yang 1 (YY1) in Leishmania infection. The manuscript draws mainly from recent work, especially the study by Brar et al. (2025), and offers a clear picture of YY1 biological functions, how the parasite may exploit it, and its potential role in supporting parasite survival within host cells. The writing is generally clear and precise, and the content is thoughtfully structured. References are appropriate and well integrated, helping to support the discussion throughout. The figures are useful, and the section on open questions adds value by pointing to future research directions. The manuscript flows well and provides a solid contribution to the field of host–parasite interactions. I have no specific corrections...
Author Response
Concerns: None
Answer: We would like to thank this reviewer for his/her time to evaluate our manuscript critically and for finding this review a solid contribution to the field of host-pathogen interactions.